# Dorsal and median raphe neuronal firing dynamics characterized by nonlinear measures

**Claudia Pascovich** [1,2]*, **Diego Serantes** [1], **Alejo Rodriguez** [1], **Diego Mateos** [3,4,5], **Joaquín González** [1], **Diego Gallo** [1], **Mayda Rivas** [1], **Andrea Devera** [1], **Patricia Lagos** [6], **Nicolás Rubido** [7‡]*, **Pablo Torterolo** [1‡]

**1** Laboratory of Sleep Neurobiology, Department of Physiology, Facultad de Medicina, Universidad de la República, Montevideo, Uruguay, **2** Consciousness and Cognition Laboratory, Department of Psychology, King's College, University of Cambridge, Cambridge, United Kingdom, **3** Achucarro Basque Center for Neuroscience, Leioa (Bizkaia), Spain, **4** Instituto de Matemática Aplicada del Litoral (IMAL-CONICET-UNL), Santa Fé, Argentina, **5** Universidad Autónoma de Entre Ríos (UADER), Oro Verde, Entre Ríos, Argentina, **6** Laboratory of Neuropeptide Transmission, Department of Physiology, Facultad de Medicina, Universidad de la República, Montevideo, Uruguay, **7** Institute for Complex Systems and Mathematical Biology, King's College, University of Aberdeen, Aberdeen, United Kingdom

‡ These authors share senior authorship on this work.
* cpascovich@gmail.com (CP); nicolas.rubidoobrer@abdn.ac.uk (NR)

**Data Availability Statement:** All relevant data are within the paper, its Supporting information files and the code for computation of classic and nonlinear measures used for analysis is available in

## Abstract

The dorsal (DRN) and median (MRN) raphe are important nuclei involved in similar functions, including mood and sleep, but playing distinct roles. These nuclei have a different composition of neuronal types and set of neuronal connections, which among other factors, determine their neuronal dynamics. Most works characterize the neuronal dynamics using classic measures, such as using the average spiking frequency (FR), the coefficient of variation (CV), and action potential duration (APD). In the current study, to refine the characterization of neuronal firing profiles, we examined the neurons within the raphe nuclei. Through the utilization of nonlinear measures, our objective was to discern the redundancy and complementarity of these measures, particularly in comparison with classic methods. To do this, we analyzed the neuronal basal firing profile in both nuclei of urethane-anesthetized rats using the Shannon entropy (Bins Entropy) of the inter-spike intervals, permutation entropy of ordinal patterns (OP Entropy), and Permutation Lempel-Ziv Complexity (PLZC). Firstly, we found that classic (i.e., FR, CV, and APD) and nonlinear measures fail to distinguish between the dynamics of DRN and MRN neurons, except for the OP Entropy. We also found strong relationships between measures, including the CV with FR, CV with Bins entropy, and FR with PLZC, which imply redundant information. However, APD and OP Entropy have either a weak or no relationship with the rest of the measures tested, suggesting that they provide complementary information to the characterization of the neuronal firing profiles. Secondly, we studied how these measures are affected by the oscillatory properties of the firing patterns, including rhythmicity, bursting patterns, and clock-like behavior. We found that all measures are sensitive to rhythmicity, except for the OP Entropy. Overall, our work highlights OP Entropy as a powerful and useful quantity for the characterization of neuronal discharge patterns.

GitHub at the following link: https://github.com/Cpasco/Entropy_Raphe.

**Funding:** This study was supported by the "Programa de Desarrollo de Ciencias Básicas, PEDECIBA" (https://www.pedeciba.edu.uy/es/area/biologia/) and the "Comisión Sectorial de Investigación Científica" (CSIC) I + D grupos 2022-22620220100148 grant from Uruguay (https://www.csic.edu.uy/). The funders had no role in study design, data collection and analysis, decision to publish, or preparation of the manuscript.

**Competing interests:** The authors have declared that no conflict of interests exist.

## Author summary

The spiking activity of neurons in a certain population can be characterized by different measures. Classic methods to do this characterization are the firing rate, which is the number of spikes per second, the action potential duration, which measures the duration of the spike, and the coefficient of variation, which quantifies the regularity in the inter-spike intervals. In this work, we tested these linear methods and added three different measures of complexity to characterize the neuronal activity of two groups of neurons in the rat: the dorsal and median raphe. These nonlinear measures quantify the degree of uncertainty or average information in the sequence of inter-spike intervals, and are called Shannon entropy (Bins Entropy), Permutation entropy (OP Entropy), and Permutation Lempel-Ziv Complexity (PLZC). We found that only the OP Entropy was sensitive to the different activity from the DRN and MRN neurons. Additionally, OP Entropy has either a weak or no relationship with the rest of the measures tested, providing complementary information to the characterization of the neuronal activity. We also show that OP Entropy is unaffected by other patterns in the signal, like rhythmicity or burst patterns, concluding that OP Entropy is a good measure to characterize the neuronal activity.

## 1 Introduction

The raphe nuclei are located in the brain stem along its mid-line [1]. A large body of evidence supports the role of the DRN (dorsal raphe nucleus) and MRN (median raphe nucleus) in an extensive array of important functions, including stress response [2–5], pain control [6–9], reproductive functions and behavior [10], food intake and obesity [11, 12], aggressiveness [13, 14], social interaction [15], motivation and reward [16, 17], fear [18, 19], learning and memory [20–23], motor activity [12], and the sleep-wake cycle physiology [24–26]. Additionally, the raphe nuclei play an important role in the physiopathology of several diseases, including major depression [27–29], anxiety, panic, obsessive compulsive disorder, eating disorders, phobias, drug addiction, and post-traumatic stress [30].

Despite the raphe nuclei participating in similar physiological functions, the DRN and MRN play distinct roles in neurotransmission and physiological functions. For example, the microinfusion of a selective agonist of the 5-HT1A receptor (8-OH DPAT) into the MRN inhibits the activity of serotonergic neurons, which produces general behavioral hyperactivity; contrary to what happens in the DRN with a decrease in certain behaviors, such as rearing and grooming [12]. Also, lesions to the DRN increase reactivity and aggression evoked by pain, while lesions to the MRN do not have such effect [6, 9]. Furthermore, both nuclei comprise varying proportions of the same neurotransmitters, thereby influencing distinct roles. The neuronal dynamics also depends on other local factors, including its connections; these nuclei have different projections, and in turn, receive afferents from different regions. A comparison of the DRN and MRN projections shows they are distributed in regions that, in most cases, do not overlap in the anterior brain; instead, they project to complementary regions. For example, the DRN projects to the medial prefrontal cortex, the amygdala, and the accumbens nucleus, whereas the MRN projects to medial structures, including the medial septum and diagonal band, zona *incerta*, and posterior hypothalamus [30, 31]. In summary, the differentiation between DRN and MRN neurons is crucial for a more nuanced understanding of their roles in neurobiology, behavior, and various physiological functions.

To date, most studies characterizing neuronal groups using electrophysiological extracellular recordings focus on: the frequency of neuronal discharge, the rhythmicity (measured by

the auto-correlation of events), the regularity (measured by the coefficient of variation), and the phase coherence (or phase locking) with hippocampal, cortical or other biological rhythms [32–40]. For example, serotonergic neurons from the DRN have been described as clock-like, exhibiting spontaneous activity, a slow $1 - 5$ $Hz$ firing-rate [41], regular spiking activity (coefficients of variation close to zero), wide action potentials ($>1.4$ $ms$), and to have rhythmic patterns of discharge [41, 42], suggesting a "neuronal signature" of this neuro-chemical group. However, there are serotonergic neurons with different electrophysiological properties as well as non-serotonergic neurons in the DRN. In addition, neurons from the MRN have heterogeneous firing patterns [39, 43–46]. Therefore, there is a need to find new characteristics in the neuronal recordings in order to better differentiate between different neuronal groups, and in the case of the present paper, between DRN and MRN neurons.

The classical characterization of neurons mainly relies on analyzing the salient characteristics of the distribution of inter-spike intervals (such as the distribution's mode and coefficient of variation) and applying linear methods (such as the auto-correlation or Z-coherence), which miss the nonlinear components that are naturally present in the neuronal dynamics. Moreover, some reports describe neuronal characteristics with redundant measures; for example, mean firing-rate and mean inter-spike interval (being the inverse of each other), or the auto-correlation and the spectral components (being related by their Fourier transform), which provide no new information about the spiking characteristics.

Nonlinear methods can address this problem and may reveal complementary information from the recordings, which has been the case for the electroencephalogram (EEG) [47–50]. So, we propose to use nonlinear methods to extend the characterization of neuronal activity. To achieve this objective, our present study involved analyzing the firing profiles (the inter-spike intervals) of individual neurons in both the DRN and MRN. We computed the entropy from the time-series distribution of amplitudes (i.e., Bin Entropy) [51], Permutation Entropy [52], and Permutation Lempel-Ziv complexity [53], comparing them with classic electrophysiological methods, which include the average firing frequency rate (FR), action potential duration (APD), and coefficient of variation (CV). We employed nonlinear measures and compared them with the classic measures to discern potential redundancy in the information obtained. Additionally, we assessed their sensitivity to the oscillatory properties of the firing patterns, encompassing rhythmicity, bursting patterns, and clock-like behavior.

## 2 Results

We recorded a total of 169 neurons from both nuclei, corresponding 77 to the DRN and 92 to MRN of the rat. We confirmed that the neurons were located within the limits of the corresponding nuclei by reconstruction of the micropipette tracts, or by identification of the recorded neuron with neurobiotin (Nb). Fig 1 shows examples of the recording, processing, and recognition of Nb-labeled neurons.

The entire population of recorded neurons within the DRN ($n = 77$) display the following electrophysiological characteristics. An APD of $3.07 \pm 0.37$ $ms$ (mean ± SEM), FR $8.45 \pm 1.94$ $Hz$, and CV $0.65 \pm 0.079$. 79% percent of these neurons exhibit uni-modal interval histogram (IH), 26% have a rhythmic pattern of discharge in the ACH, and 20% have a predominant interval in the auto-correlation histogram (ACH). Moreover, a burst firing pattern is observed in 4 neurons (5%), i.e., showing doublets or triplets with $< 20$ $ms$ intervals and a prominent decrease in the amplitude of higher order spikes [32]. On the other hand, the recorded neurons in the MRN ($n = 92$) display the following electrophysiological characteristics. An APD of $2.56 \pm 0.27$ $ms$, FR of $10.52 \pm 1.34$ $Hz$, and CV of $0.76 \pm 0.055$. 75% of these neurons exhibit

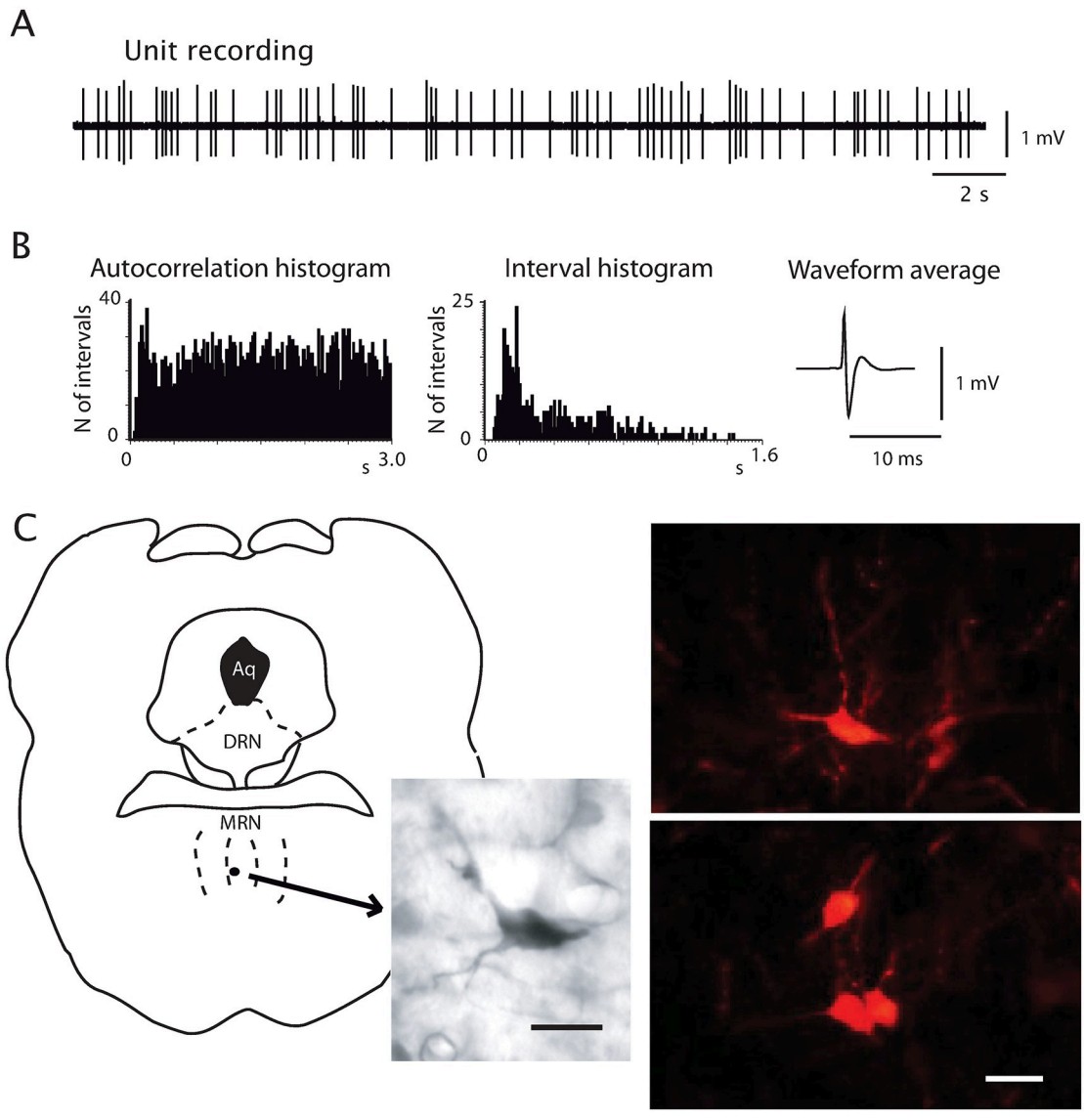

**Fig 1. Neuronal recording and identification example.** A) Raw recording of a MRN neuron. B) Autocorrelation histogram, interval histogram, and average waveform for the same neuron. C) Examples of neurons identified by neurobiotin. Antero-posterior coordinate from Bregma -8 mm according to [54]. Calibration bars: 20 $\mu$m.

uni-modal IH, 21% a rhythmic pattern, and 35% a predominant interval in the ACH. Burst firing pattern is observed in 20 neurons (21%).

## 2.1 OP Permutation Entropy distinguishes DRN and MRN neuronal activity

We first studied whether the nonlinear measures—OP Entropy, PLZC, and Bins Entropy—can distinguish MRN from DRN neuronal dynamics compared to the classic measures—APD, CV, and FR.

For the classic measures, it can be seen from Fig 2 that only the CV has just a tendency to discriminate both nuclei (0.77 ± 0.05 for the MRN Vs. 0.61 ± 0.08 for the DRN, with a non-

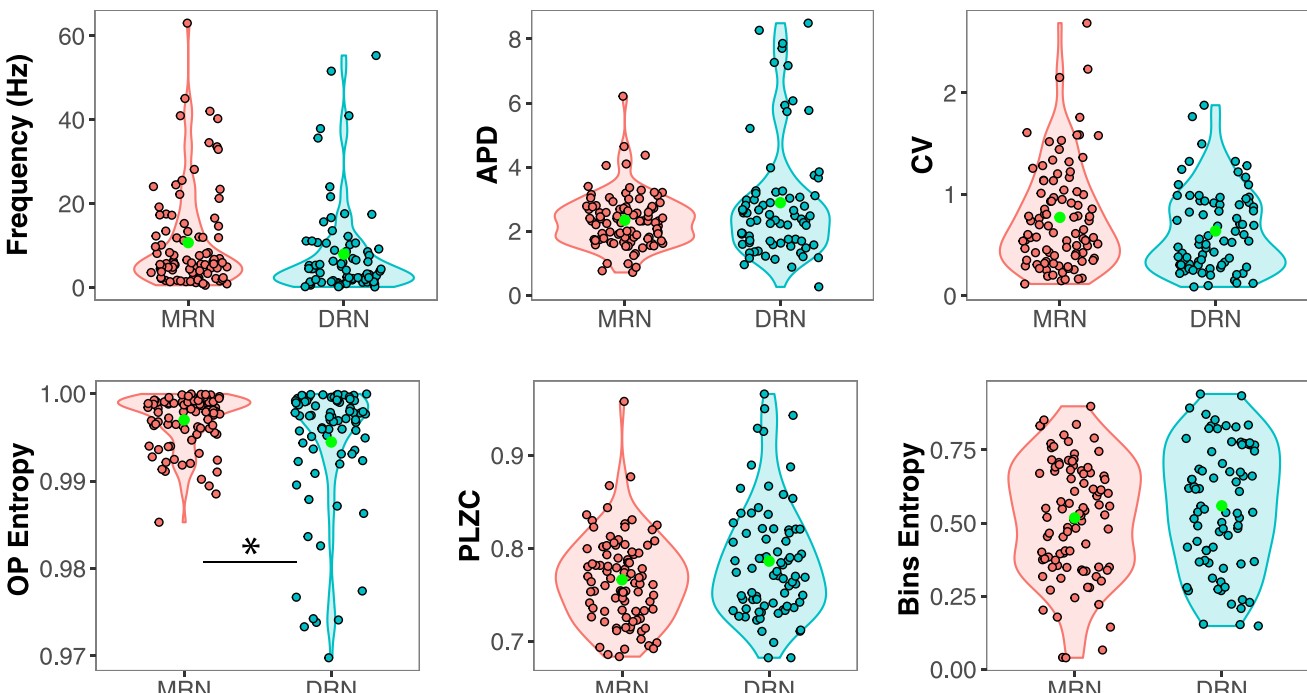

**Fig 2. Comparison between classic measures (top) and nonlinear measures (bottom) from MRN and DRN neuronal sub-populations.** Classic measures (FR, firing frequency rate; APD, action potential duration; CV, coefficient of variation) are insensitive to the neuronal activity differences in the nuclei, and only the permutation entropy (OP Entropy) among the nonlinear measures (PLZC, Permutation Lempel-Ziv complexity; Bins Entropy, Shannon entropy of histogram of inter-spike intervals using 18 bins) is significantly different between both neuronal groups: $^{*}p < 0.05$.

significant $p = 0.06$). The FR and APD are not significantly different between groups ($10.74 \pm 1.39$ Vs. $8.72 \pm 2.08$, $p = 0.30$; and $2.47 \pm 0.29$ Vs. $2.72 \pm 0.40$ $ms$, $p = 0.33$, respectively). For the nonlinear measures, only the OP Entropy is significantly lower for the neurons from the DRN than for those from the MRN ($0.9970 \pm 5.714 \times 10^{-4}$ Vs. $0.9954 \pm 8.460 \times 10^{-4}$, $p < 0.05$), which could be attributed to a sub-population of neurons with an OP Entropy lower than 0.985. Bins Entropy and PLZC show non-significant differences between the values from both groups ($0.52 \pm 0.02$ Vs. $0.54 \pm 0.03$, $p = 0.63$; and $0.76 \pm 0.005$ Vs. $0.77 \pm 0.008$, $p = 0.28$, respectively). Moreover, OP Entropy and PLZC values are significantly different than those obtained from surrogates of the ISI sequences (for more information see S1 Fig).

We also explored different paired combinations of the measured variables and visualized the resultant scatter plots. For each pair of measures, two shaded contours are plotted surrounding the data-points belonging to the DRN (light blue) and MRN (orange). By doing this, we can qualitatively see if the DRN and MRN neuronal populations have superimposed values or the measures hold mostly different values (Fig 3).

Among the pairwise comparison of classic measures, the combination of FR and CV shows that the values are similar for both nuclei (Fig 3A). The combination of APD with FR or CV only partially separates both nuclei, where the DRN shows a group with high APD and low FR (Fig 3B), and the MRN a group with low APD and high CV (Fig 3C).

When comparing the APD with nonlinear measures (Fig 3D, 3E and 3F), the combinations show that some DRN neurons (highlighted by the shaded cyan area) have different values than those from the superimposed MRN neurons (highlighted by the shaded orange area). For example, it can be seen (Fig 3D) that neurons with OP Entropy lower than 0.985 have a short APD and that these neurons belong to the DRN. By contrast, MRN neurons show OP Entropy

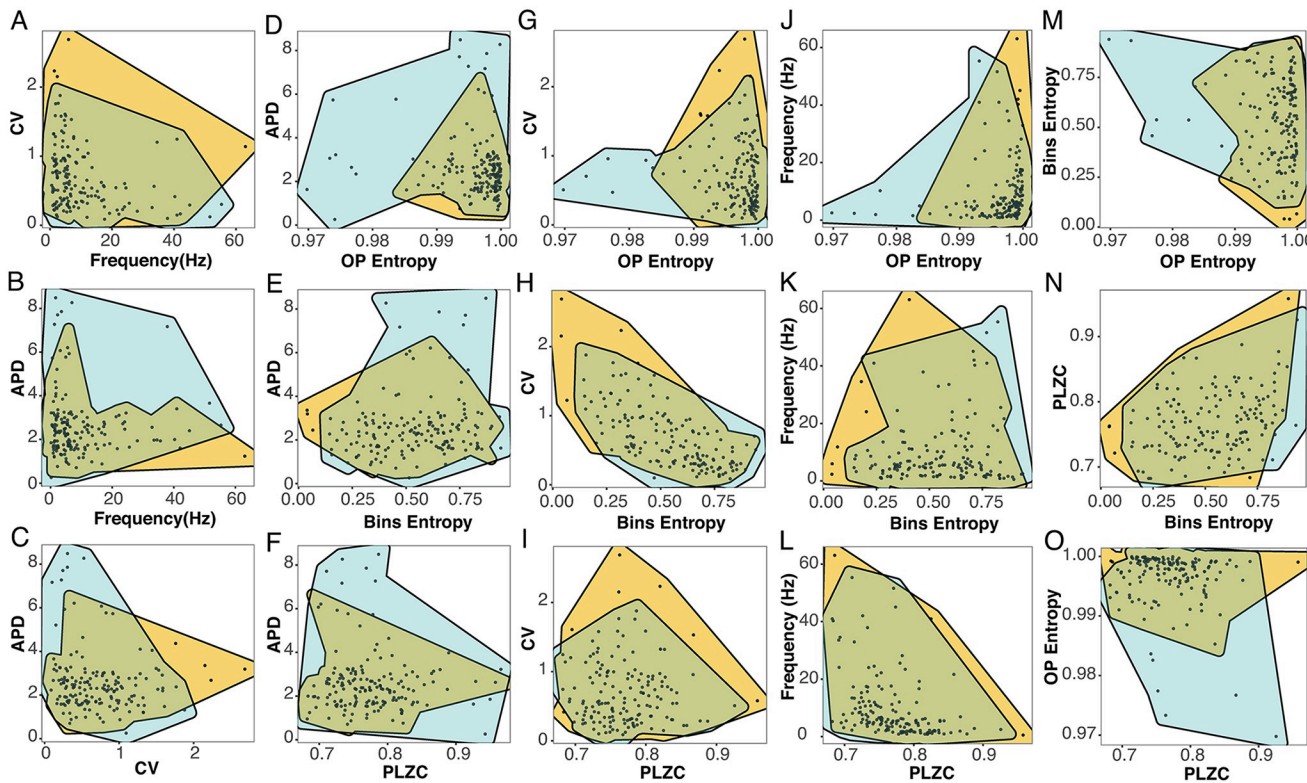

**Fig 3. Scatter plots of values from the linear and nonlinear measures for the DRN (shaded cyan areas) and MRN (shaded orange areas).** The plotted pairwise measures are those from Fig 2.

higher than 0.985. For Bins Entropy and PLZC the non-superimposed area predominates in the upper part of the plots, for the neurons with larger APD from the DRN (Fig 3E and 3F).

When looking at the relationship between the CV and nonlinear measures (Fig 3G, 3H and 3I), results vary. The combination of CV and OP Entropy can distinguish some DRN from MRN neurons (Fig 3G). There is a group of neurons with OP Entropy lower than 0.985 that purely contain DRN neurons with small CV, whereas for higher values of OP Entropy both neuronal populations are mixed (as in Fig 2G). On the other hand, we cannot see differences between DRN and MRN neurons when looking at the combination of CV and Bins Entropy or PLZC (Fig 3H and 3I). Similarly, the combination of FR with nonlinear measures (Fig 3J, 3K and 3L) shows partial separations between the nuclei only when combined with OP Entropy (Fig 3J), where OP Entropy values below 0.98 contain neurons from the DRN with low FR.

Finally, the combinations between the entropies (Fig 3M, 3N and 3O) show separate characteristics for some of the neurons from the DRN and MRN populations. When using OP Entropy and Bins Entropy (Fig 3M), a light blue area can be seen containing neurons with low OP Entropy and high Bins Entropy; and when using PLZC with OP Entropy, DRN neurons with low OP Entropy show a range of PLZC values (Fig 3O).

From these results, we can conclude that the APD combined with nonlinear measures (Fig 3D, 3E and 3F) are likely to provide complementary information to characteize the two raphe neuronal populations. Additionally, the OP Entropy can also separate the discharge pattern of these neuronal populations in combination with other variables, like the CV, FR, Bins Entropy, and PLZC.

## 2.2 Relationship between classic and nonlinear measures

In order to explore in more detail the relationship between the different measures for both nuclei, we plotted all possible paired combinations of the variables and studied the relationship between them (Fig 4 and Table 1), by the use of multilevel linear model fitting. In some cases, a log scale transformation was performed before fitting. When a linear model significantly fits in a log-log scale transformed combination of variables, the relation was considered nonlinear.

For the classic measures, there is a statistically significant negative nonlinear relationship between the FR and the CV ($p < 0.001$, Fig 4A), but there is neither a linear relationship between the FR and APD, nor between the CV and the APD ($p = 0.61$ in Fig 4B and $p = 0.14$ in Fig 4C).

When comparing classic to nonlinear measures, we can see significant ($p < 0.001$) nonlinear relationships between the CV and Bins Entropy (Fig 4H) and between the FR and PLZC (Fig 4L). With less statistical significance ($p < 0.05$), we also find nonlinear relationships between the APD and OP Entropy (Fig 4D) and between the FR and OP Entropy (Fig 4J) The exponents for these nonlinear relationships are $\beta_H = -1.59$ [i.e., CV $\propto$ (bin entropy)$^{\beta_H}$], $\beta_L = -9.27$ [i.e., FR $\propto$ (PLZC)$^{\beta_L}$], $\beta_D = 0.46$ [i.e., APD $\propto$ (1 $-$ OP entropy)$^{\beta_D}$], and $\beta_J = -53.55$ [i.e., FR $\propto$ (1 $-$ OP entropy)$^{\beta_J}$]. We cannot see other linear or nonlinear relationships between the classic and nonlinear measures, which is summarized in Table 1.

Regarding the nonlinear measures, we found significant linear relationships between them. It can be seen that the Bins Entropy and OP Entropy have a weak linear relationship ($p = 0.024$, $r_M = 0.17$; Fig 4M), similar to the one between OP Entropy and PLZC ($p = 0.022$,

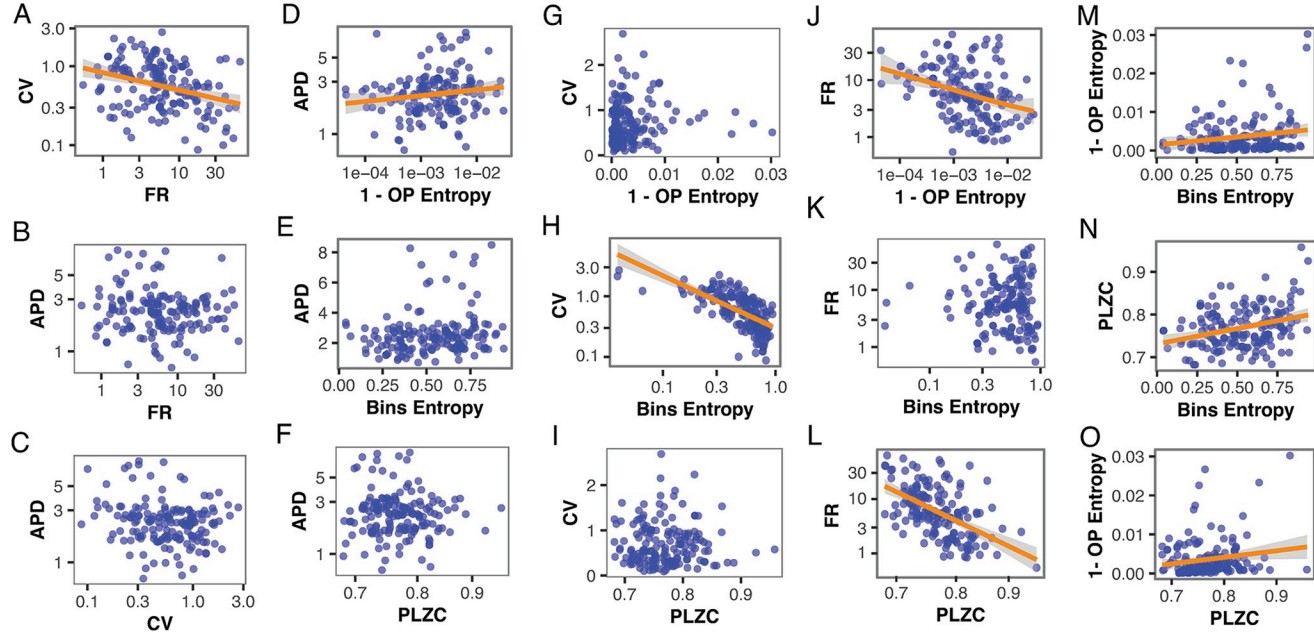

**Fig 4. Pairwise measures used to characterize neuronal dynamics from inter-spike intervals coming from 169 neurons of the raphe nuclei.** The best fit for the values in panel A happens in log-log scale, meaning that a nonlinear relationship exists between CV and FR, with an exponent $\beta_A = -0.50$ [i.e., CV proportional to FR$^{\beta_A}$] and a Pearson correlation coefficient $r_A = -0.32$. Similarly, the best fit for the values in panels D, H, J, and L also happens in the log-log scale, where the respective exponents are $\beta_D = 0.46$ [i.e., APD $\propto$ (1 $-$ OP entropy)$^{\beta_D}$] with $r_D = 0.18$, $\beta_H = -1.59$ [i.e., CV $\propto$ (bin entropy)$^{\beta_H}$] with $r_H = -0.65$, $\beta_J = -53.55$ [i.e., FR $\propto$ (1 $-$ OP entropy)$^{\beta_J}$] with $r_J = -0.25$, and $\beta_L = -9.27$ [i.e., FR $\propto$ (PLZC)$^{\beta_L}$] and $r_L = -0.54$. Contrary, the best fit for panels M, N, and O happens in a linear scale, meaning linear relationships with slopes given by $\alpha_M = 4.11 \times 10^{-3}$ and $r_M = 0.17$, $\alpha_N = 0.07$ and $r_N = 0.31$, and $\alpha_O = 0.02$ and $r_O = 0.16$, respectively.

**Table 1. Statistical comparison between measures using linear multilevel models.** F statistic t-test is used to test the relationship between the fixed effects. $^*p < .05$, $^{**}p < .005$, $^{***}p < .001$.

|  | Mean Freq. | CV | APD | OP Entropy | Bins Entropy | PLZC |
|---|---|---|---|---|---|---|
| Mean Freq. |  | $< 0.001^{***}$ | 0.338 | $0.001^{**}$ | 0.864 | $< 0.001^{***}$ |
| CV | $< 0.001^{***}$ |  | 0.136 | 0.132 | $< 0.001^{***}$ | 0.909 |
| APD | 0.338 | 0.136 |  | $0.0192^*$ | 0.169 | 0.427 |
| OP Entropy | $0.001^{**}$ | 0.132 | $0.0192^*$ |  | $0.024^*$ | $0.035^*$ |
| Bins Entropy | 0.864 | $< 0.001^{***}$ | 0.169 | $0.024^*$ |  | $< 0.001^{***}$ |
| PLZC | $< 0.001^{***}$ | 0.909 | 0.427 | $0.035^*$ | $< 0.001^{***}$ |  |

$r_O = 0.16$; Fig 4O). By contrast, PLZC and Bins Entropy are positively correlated ($p < 0.001$, $r_N = 0.31$; Fig 4N).

In Table 1 we summarize the statistical results for all the linear combinations of the classic and nonlinear measures.

## 2.3 Sensitivity of the classic and nonlinear measures to rhythmic pattern

Next, we examined the impact of neuronal firing rhythmicity on both classic and nonlinear measures. Specifically, certain neurons in the DRN and MRN exhibit predominant firing frequencies within the range of 1 up to 60 *Hz*, as illustrated in Fig 5A. These neurons display periodic firing patterns and exhibit an ACH characterized by regular peaks, as depicted in Fig 5B. In the MRN, neuronal firing predominantly exhibits a theta rhythm ($4 - 9$ *Hz*), evident in the ACH. Meanwhile, in the DRN, a lower frequency prevails (Fig 5A). These results are in line with previous evidence showing that a subgroup of neurons, discharged rhythmically in synchrony with the hippocampal theta rhythm; synchronization of the firing with the hippocampal theta rhythm is greater in the MRN than in the DRN [37].

We classified the neurons from each nucleus into two categories: rhythmic and non-rhythmic. As a result, we got 19 rhythmic neurons (out of 92: 21%) and 40 non-rhythmic neurons (out of 92: 44%) in the MRN and we got 20 rhythmic neurons (out of 77: 26%) and 41 non-rhythmic neurons (out of 77: 54%) in the DRN. Neurons having an ACH without defined peaks or showing only a unique predominant interval in the ACH were excluded from this analysis, as in the example of Fig 5D (33 in the MRN and 16 in the DRN). Hence, we conducted a comparison between distinctly rhythmic patterns, with multiple discernible peaks in the ACH (Fig 5B), and patterns that were clearly non-rhythmic (Fig 5C). Then, we tested if the different measures are affected by rhythmic properties and explored the ability of each of the measures in separating DRN and MRN within rhythmic and non-rhythmic subgroups.

When looking at the classic measures, it can be seen from Fig 5E that the FR, APD, and CV are modulated by rhythmicity depending on whether the neurons are from the MRN or DRN. Rhythmic neurons from the MRN group have a significantly higher FR than non-rhythmic ones ($16.96 \pm 3.07$ *Hz* and $7.43 \pm 1.77$ *Hz* respectively, $p = 0.008$), but rhythmic and non-rhythmic neurons from the DRN have similar FR values ($12.60 \pm 3.49$ *Hz* and $8.07 \pm 2.11$ *Hz* respectively, $p = 0.23$). On the other hand, APD is significantly larger in rhythmic neurons from the DRN than non-rhythmic ones ($3.89 \pm 0.50$ *ms* Vs. $2.4661 \pm 0.29$ *ms*, respectively, $p = 0.005$), but the APD shows no significant differences in the rhythmicity of MRN neurons ($2.23 \pm 0.12$ *ms* Vs. $2.23 \pm 0.21$ *ms*, $p = 0.998$). Similarly, rhythmic neurons from the DRN have significantly smaller CV than non-rhythmic ones ($0.46 \pm 0.11$ and $0.74 \pm 0.06$ respectively, $p = 0.01$), but are undifferentiated in the MRN ($0.71 \pm 0.12$ Vs. $0.78 \pm 0.073$, $p = 0.54$).

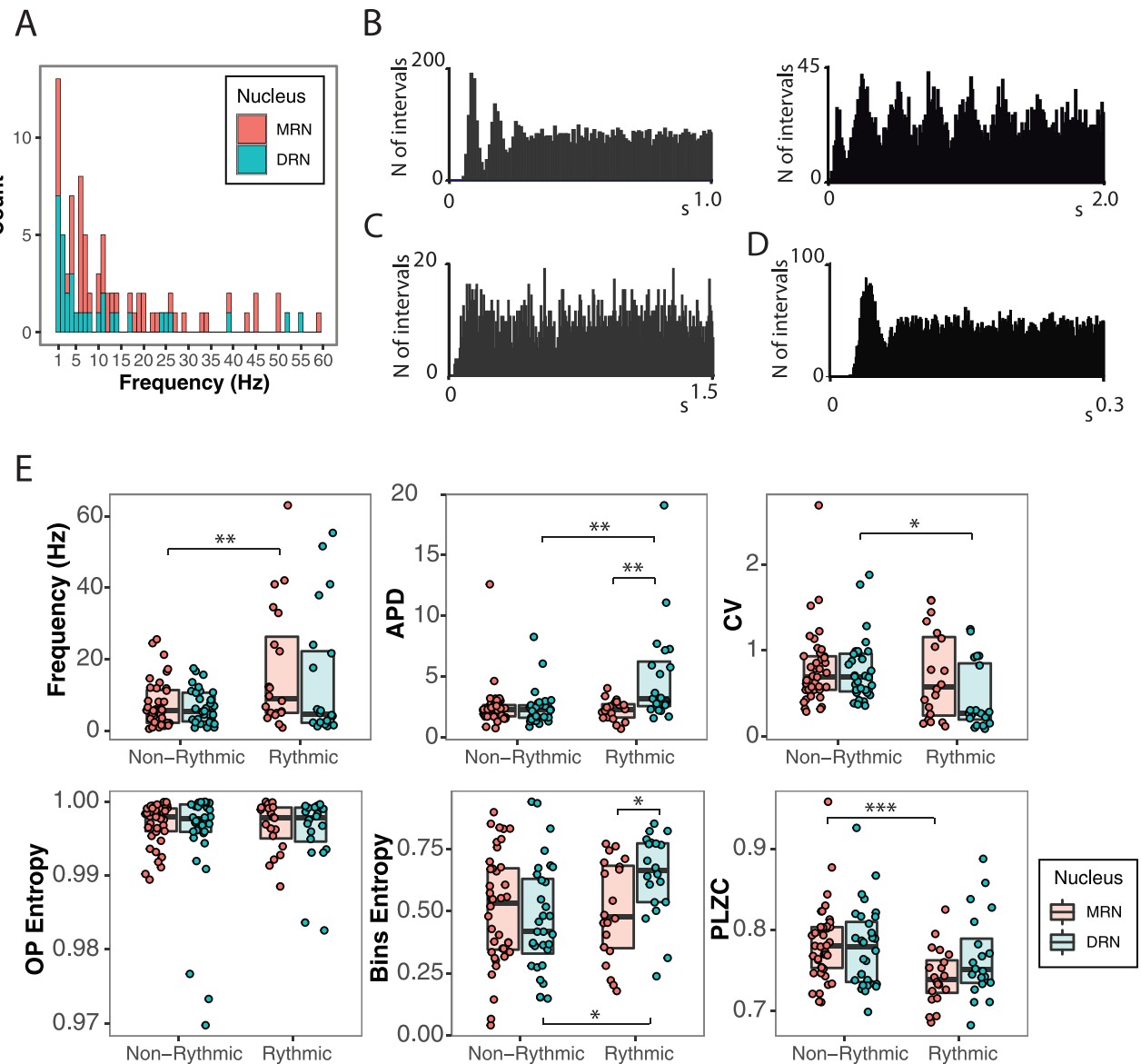

**Fig 5. Sensitivity of the classic and nonlinear measures to rhythmic patterns in the neuronal discharge.** A) Distribution of the frequencies of the rhythmic neurons of the DRN and MRN. B) ACH of rhythmic neurons. C) ACH of a non-rhythmic neuron. D) Example of a single predominant interval in the ACH. E) Relationship between classic and nonlinear measures with rhythmicity, where OP Entropy is the only measure insensitive to the rhythmic pattern. The median is represented with horizontal black lines. *p < .05, **p < .005, ***p < .001.

Regarding the nonlinear measures, it can be observed from Fig 5E that OP Entropy is the only one insensitive to rhythmicity. For the DRN, the OP Entropy for non-rhythmic and rhythmic neurons is 0.99 ± 0.0011 and 0.99 ± 0.0019 ($p = 0.77$), respectively. For the MRN, the OP Entropy values are 0.99 ± 0.00048 and 0.99 ± 0.00082, respectively.

On the other hand, Bins Entropy significantly increases to 0.63 ± 0.059 in the rhythmic neurons of the DRN versus the non-rhythmic neurons, which hold 0.47 ± 0.035 ($p = 0.006$). However, no differences are found for the MRN, with Bins Entropies of 0.50 ± 0.034 and 0.50 ± 0.059 ($p = 0.956$) for non-rhythmic and rhythmic neurons, respectively. Similarly, PLZC is sensitive to rhythmic properties of the neuronal discharge, where rhythmic neurons

have lower PLZC of 0.74 ± 0.011 than non-rhythmic neurons, with 0.78 ± 0.0068. However, it was only statistically significant for the MRN ($p < 0.001$). No differences were found for the DRN (0.77 ± 0.0088 Vs. 0.765 ± 0.012, $p = 0.64$).

When looking at the ability of linear and nonlinear measures to differentiate between DRN and MRN neurons within rhythmic or non-rhythmic subgroups, it can be seen that only APD and Bins Entropy achieve a significant difference in their values for rhythmic neurons. APD is larger for the DRN (0.66 ± 0.058 *ms*) compared to the MRN (0.50 ± 0.039 *ms*, $p = 0.012$), and Bins Entropy is higher for the DRN (0.50 ± 0.041 Vs. 0.63 ± 0.059, $p = 0.04$) compared to the MRN. However, no measures separates the nuclei when looking at the non-rhythmic subgroup of neurons.

## 2.4 Nonlinear measures are insensitive to burst patterns

Some neurons in DRN (5%) and MRN (21%) display a burst firing pattern. An example of this firing pattern is shown in Fig 6A. In order to study how this pattern of activity can affect the

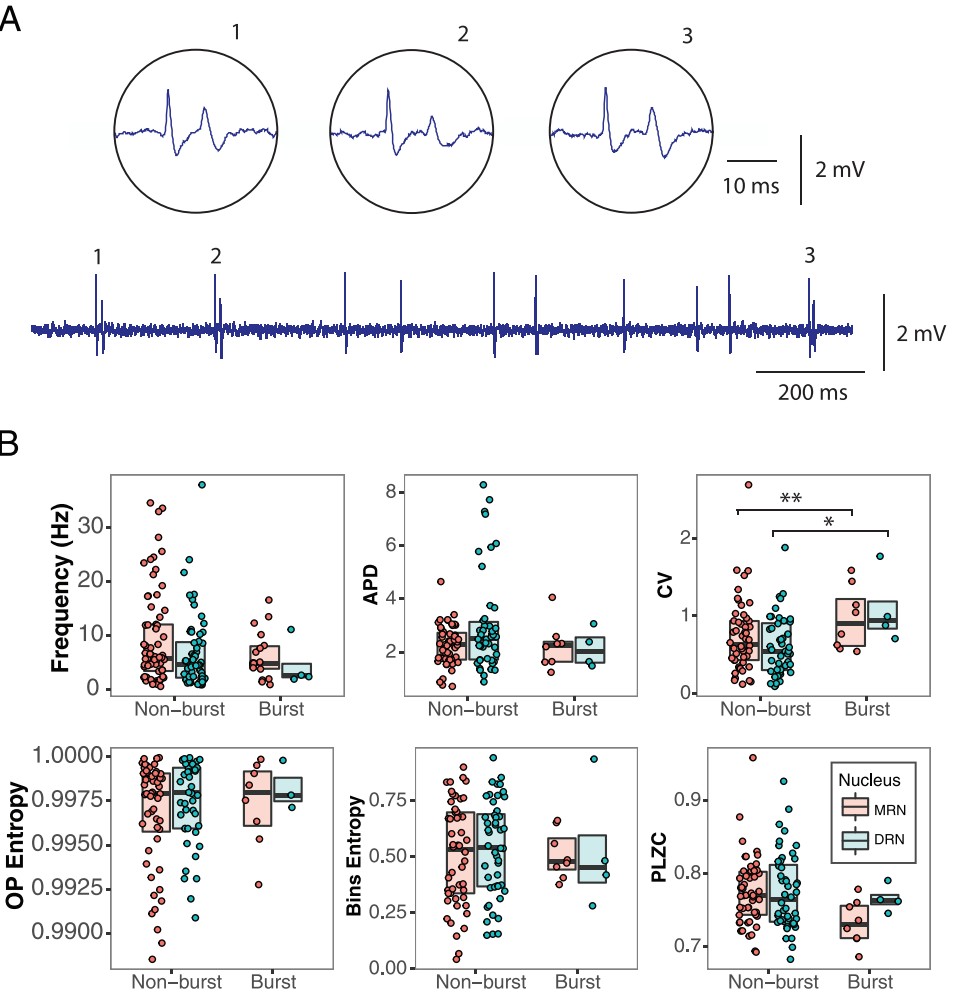

**Fig 6. Sensitivity of the classic and nonlinear measures to the presence of bursts in the neuronal discharge.** A) Example of a raw neuronal recording with bursts (indicated with numbers). A zoomed view of these bursts is shown in the circular boxes. B) Comparison of the six measures studied on their sensitivity to bursts—nonlinear measures appear insensitive to bursts. The CV is the only measure that shows a higher value for the bursting neurons in both nuclei. The median is represented by horizontal black lines. *p < .05, **p < .005, ***p < .001.

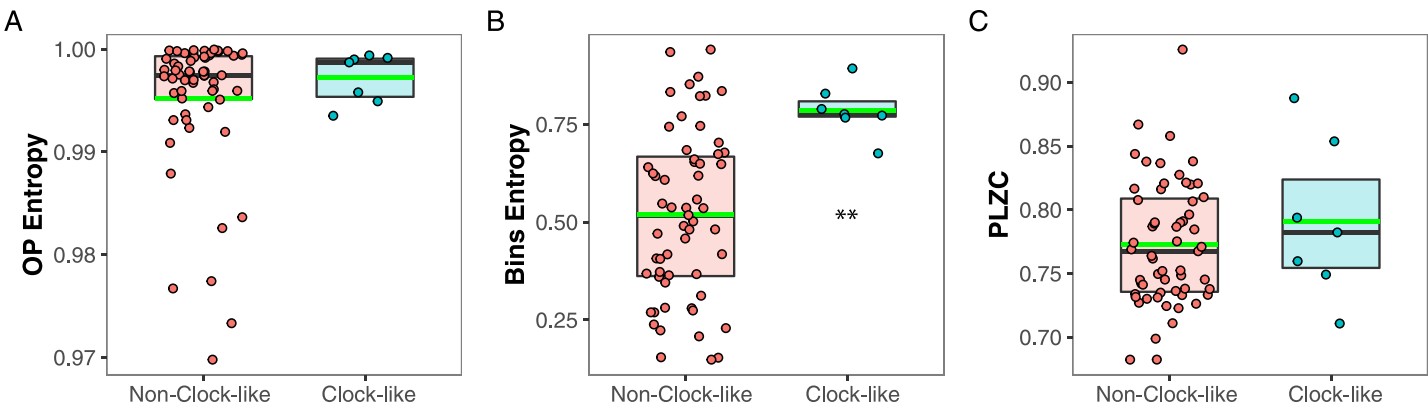

**Fig 7. Characterization of the clock-like and non-clock-like neurons using nonlinear measures: (A) OP Entropy, (B) Bins Entropy and (C) PLZC.** Clock-like neurons show high Bins Entropy whereas non-clock-like sub-group shows a widespread distribution. OP, Ordinal Patterns; PLZC, Permutation Lempel-Ziv complexity. The median is represented by horizontal black lines and the mean with green lines. **p < .005.

classic and nonlinear measures we compared neurons with and without burst pattern activity for each of the variables and for both nuclei (Fig 6B). We encountered that of the classic measures only the CV is sensitive to burst activity having the neurons with burst a higher CV for DRN ($1.07 \pm 0.19$ Vs. $0.58 \pm 0.05$; $p < 0.05$) as well as for the MRN ($1.07 \pm 5.31$ Vs. $0.69 \pm 1.90$; $p < 0.01$). None of the other measures are affected by the presence of burst pattern of activity (FR for DRN: $4.59 \pm 5.31$ Vs. $10.13 \pm 1.90$, $p = 0.36$; FR for MRN: $15.61 \pm 5.31$ Vs. $9.81 \pm 1.90$, $p = 0.98$; APD for DRN: $2.13 \pm 1.26$ Vs. $3.41 \pm 0.50$, p = 0.37; APD for MRN: $2.44 \pm 0.34$ Vs. $2.45 \pm 0.16$, p = 0.98; OP Entropy for DRN: $0.992 \pm 3.3 \times 10^{-3}$ Vs. $0.995 \pm 0.82 \times 10^{-3}$, p = 0.27; OP Entropy for MRN: $0.996 \pm 0.77 \times 10^{-3}$ Vs. $0.997 \pm 0.35 \times 10^{-3}$, p = 0.27; PLZC for DRN: $0.76 \pm 0.25 \times 10^{-1}$ Vs. $0.77 \pm 0.63 \times 10^{-2}$, p = 0.67; PLZC for DRN: $0.74 \pm 1.17 \times 10^{-2}$ Vs. $0.77 \pm 0.54 \times 10^{-2}$, p = 0.05; Bins Entropy for DRN: $0.9921 \pm 0.11427$ Vs. $0.54697 \pm 0.02813$, p = 0.52. Bins Entropy for MRN: $49.86 \times 10^{-2} \pm 0.50 \times 10^{-1}$ Vs. $52.31 \times 10^{-2} \pm 0.23 \times 10^{-1}$, p = 0.61).

## 2.5 A high Bins Entropy characterize clock-like neurons

Clock-like neurons had been determined to have APD larger than 1.4, CV lower than 0.30, rhythmic pattern in the ACH, and low FR ($< 5\ Hz$), and this type of neuron have also been shown to be serotonergic [34, 36, 55, 56]. In this work, neurons considered as clock-like were separated from the non clock-like following the previous characteristics, and their spiking activity were compared using nonlinear measures. We found 7 neurons with the aforementioned clock-like characteristics only in the DRN (which had 77 neurons).

As shown in Fig 7, clock-like neurons showed a higher Bins Entropy compared with the non-clock-like group ($0.51 \pm 0.03$ Vs. $0.78 \pm 0.07$, $p = 0.002$). The OP Entropy and PLZC were not statistically different between those groups (OP Entropy: $0.995 \pm 8.45 \times 10^{-4}$ Vs. $0.997 \pm 2.59 \times 10^{-3}$, $p = 0.42$; PLZC: $0.77 \pm 0.0064$ Vs. $0.79 \pm 0.019$, $p = 0.38$).

## 3 Discussion

In the present work, we characterize the dynamics of neuronal activity from the DRN (dorsal raphe nucleus) and MRN (median raphe nucleus) using 3 classic measures—FR, APD and CV—and 3 more recently developed nonlinear measures—OP Entropy, Bins Entropy and PLZC. We first summarize the evidence provided by the classical measures of the DRN and

MRN, and then we discuss the contribution of the nonlinear measures to the characterization of raphe neuronal dynamics.

## 3.1 Classic characterization of raphe neurons

Most of the studies that contributed to the characterization of DRN neurons are focused on serotonergic neurons. Early studies detected regularly spontaneously firing neurons in the DRN in anesthetized rats using extracellular recording techniques [34, 36, 55, 56]. Serotonergic neurons of the DRN characteristically show spontaneous activity with a slow (1–5 *Hz*) and regular discharge frequency described as clock-like [55]. This unique firing pattern would serve as a "neuronal signature" for this neurochemical group of brainstem cells [1]. The extracellular action potentials (APs) of these neurons exhibit a prominent positive deflection, followed by a negative or negative/positive deflection. The first positive deflection plus the first negative deflection has a duration greater than 1.4 *ms* [33, 34].

Hajos et al. in 1995 [33] described that during their regular discharge these neurons show spike bursts (from 2 to 4 spikes) [32, 33]. In these bursts, the spikes have a short interval (range: 2.4–11.5 *ms*), and the secondary spikes show a decrease in amplitude. In addition, non-serotonin neurons with varied electrophysiological properties are also present in the DRN. This heterogeneous population of neurons show a less regular pattern of firing, with firing rates in the range of 0.1–30 *Hz* [34–36, 55].

In contrast to the DRN, few studies describe the electrophysiological characteristics of the MRN. These studies of the MRN were carried out *in vitro* [46, 57, 58] or *in vivo* using anesthetized animals [33, 37–39]. The studies carried out by Kocsis et al.(2006) [38] and Viana di Prisco (2022) [39] showed a great diversity in the MRN neurons: a group of serotonergic neurons which characteristics similar to the clock-like neurons of the DRN, a fast-firing serotonergic group of neurons rhythmic with theta, and a more heterogeneous non-serotonergic group of neurons. Therefore, the electrophysiological criteria used in previous studies to identify putative serotonergic neurons would be appropriate only as a rough preliminary classification of MRN neurons emphasizing the need for new measures for the classification of neuronal activity.

## 3.2 Nonlinear measures in neuronal characterization

Nonlinear measures had been used to compare neuronal dynamics in different neurological pathologies [59], finding that their values change in different states [60, 61] and reporting that neuronal entropy depends on the level of alertness in humans and animals. It was also recently used to compare informational complexity in spike trains across species [62]. Another study [63] compared FR with the Shannon Information Transmission Rate in the responses of lateral geniculate nucleus neurons of the cat to spatially homogeneous spots of various sizes with temporally random luminance. They found that the behavior of these two rates can differ quantitatively. This suggests that the energy used for spiking does not translate directly into the information to be transmitted. They also compared FR with Information Rates for two type of cells in the lateral geniculate nucleus: X-ON (neurons excited by light onset) and X-OFF cells (neurons excited by light offset). They found that, the FR and Information Rate for X-ON cells often behave in a completely different way, while for X-OFF cells these rates are much more highly correlated. These results suggest that for X-ON cells a more efficient "temporal code" is employed, while for X-OFF cells a straightforward "rate code" is used.

A recent work by Estarellas et al. (2020) [64], used nonlinear measures to investigate the encoding and information transmission in time series of sensory neurons. They found that depending on the frequency, specific combinations of neuron/class and coupling-type allow a

more effective encoding, or a more effective transmission of the signal. However, our work is the first study proposing nonlinear measures for neuronal characterization of the raphe nuclei using novel informational complexity measures.

We show that compared to the other classic and nonlinear measures, the OP Entropy is the only measure that characterizes differently the inter-spike intervals from the DRN and MRN neuronal groups (Fig 2), pointing to OP Entropy being the strongest candidate for neuronal activity categorization.

## 3.3 Neuronal characterization by multiple complementary measures

When performing a neuronal characterization, we need to capture as much information of the spiking activity as possible, without providing redundant or overlapping information. A measure that captures the same information as another measure is redundant and not very informative.

To assess the level of redundancy among the 6 measures, we made 15 pairwise comparisons (Fig 3) and analyzed whether a functional relationship between the measures could be found (Fig 4 and Table 1). Specifically, we sought for the best fit out of linear and power-law relationships between the pairs of measures. Our results show that when we look at the classic measures that are normally used to analyze extracellular neuronal activity, the APD gives complementary information when compared to the FR or CV (Fig 4B and 4C), which means that the use of APD with either FR or CV improves the characterization of the spiking activity. Moreover, we find that FR and CV are nonlinearly related (Fig 4A), making their values redundant.

When we look at the possible redundancies between the classic and nonlinear measures, we find 4 nonlinear relationships out of the 9 possible pairs of measures. On the one hand, there are weak nonlinear relationships between the APD with the OP Entropy (Fig 4D; $r = 0.18$; $p = 0.019$) and between the FR with the OP Entropy (Fig 4J; $r = -0.25$; $p = 0.001$). This means that it is preferable to avoid using these pairs of measures together because they can provide redundant information about the neuronal dynamics of the inter-spike intervals (18% of the variation in APD is explained by variation in OP Entropy and 25% of the variation in FR is explained by variation in OP Entropy). On the other hand, there is a strong nonlinear relationship between the CV with the Bins Entropy (Fig 4H; $r = -0.65$; $p < 0.001$) and between the FR with the PLZC (Fig 4L; $r = -0.54$; $p < 0.001$). These strong relationships imply that these measures should not be used together.

When we look at the possible redundancies between the nonlinear measures, we only find weak linear relationships between the entropy measures (Fig 4M, 4N and 4O; Table 1). Because the only relationship with a strongly significant p-value is between the Bins Entropy and PLZC (Fig 4N; $r = 0.31$; $p < 0.001$), we are confident that this is the pair of measures that provides redundant information about the neuronal dynamics.

## 3.4 Measures' sensitivity to oscillatory properties in the spiking dynamics: Rhythmicity, bursts, and clock-like behavior

We show that OP Entropy is insensitive to any of the oscillatory patterns we selected from the DRN and MRN neurons. These patterns included rhythmic and non-rhythmic behavior in each nucleus (Fig 5), neurons with and without bursts (Fig 6), and clock-like versus non-clock-like activity in the DRN (Fig 7). In particular, the OP Entropy is insensitive to the rhythmic patterns present in the time series, whereas all the other measures change depending on whether the neuronal activity is rhythmic or not (Fig 5). Additionally, none of the nonlinear measures is sensitive to burst patterns (Fig 6).

Regarding the characterization of clock-like putative serotonergic neurons using nonlinear measures, we found that these neurons display a high Bins Entropy (Fig 7B). That could be because the clock-like group was selected by a low CV, which as can be seen in Fig 4H is negatively related with the Bins Entropy. As can be seen in Fig 7A, the OP Entropy, which is the measure that was significantly different between DRN and MRN, does not separate the clock-like neurons from the rest. And also, none of those neurons are in the subgroup with low levels of OP Entropy. This means that the difference in OP Entropy levels between DRN and MRN can not be attributed to clock-like neurons. On the other hand, the measure that significantly separates the clock-like neurons is the Bins Entropy. However, this measure is not different in DRN and MRN. Additionally, when we look at the distributions they overlap, meaning that we can not use the levels of Bins Entropy as a marker for clock-like neurons.

### 3.5 OP Entropy considerations

The OP Entropy values across our results show high values (Figs 2, 3, 4, 5E, 6B and 7A), even for rhythmic or clock-like neurons (Figs 5E and 7A). We checked our results and saw that the ISI sequences, even for these rhythmic or clock-like neurons, have a strong stochastic component, which supports the resultant high entropy values.

Finally, it is important to note that all the complexity analysis performed in this work were done using $D = 3$ for the ordinal patterns (OP). We were not able to test whether the differentiation improved if $D > 3$ because we are limited in the length of the time series ($N = 225$). It is probable that a larger $D$ may reveal a stronger differentiation in OP Entropy and PLZC values for the MRN and DRN neurons, as more information would be included in each OP. Similarly, we could not analyse higher-order correlations in the ISI sequence, self-similarity, or other temporal scales by changing the OP embedding delay $\tau$ (i.e., sample the ISI values within each OP window non-consecutively). This limitation remains even if we choose to encode the ISI sequence with overlapping OPs. To overcome this limitation, we need much longer ISI sequences ($N \gg 225$).

We conclude that nonlinear measures, and specially OP Entropy, contribute significantly to enrich the characterization of raphe nuclei neurons and is a promising measure to distinguish sub-populations based in neuronal dynamics.

## 4 Methods

### 4.1 Ethics statement

All of the experimental procedures were conducted in accordance with the Guide for the Care and Use of Laboratory Animals (8th edition, National Academy Press, Washington DC, 2010) and approved by the Institutional Animal Care Commission (Exp. Nº 070153-000841-18). Institutional Ethics Committee: https://www.chea.edu.uy/node/29. Adequate measures were taken to minimize pain, discomfort or stress of the animals, and all efforts were made to use the minimal number of animals necessary to produce reliable scientific data.

### 4.2 Experimental procedures

Sixty four male Wistar rats (250–310 $gr$) were used in this study. They were obtained from URBE (Reagents and Biomodels Experimentation Unit), Facultad de Medicina, Universidad de la República. The animals were maintained with food and water available *ad libitum* and kept under controlled conditions (temperature $22 \pm 2°C$, 12-h day–night cycle, lights on at 7:00 A.M.).

## 4.3 Recording procedures

Rats were anesthetized with urethane (1.5 $gr/kg$, i.p.) and positioned in a stereotaxic frame (David Kopf Instruments, USA). Following a scalp incision, skull landmarks were visualized and coordinates were determined from Paxinos and Watson, 2008 [54]. A small hole was drilled in the skull for unit recording of DRN and MRN neurons (AP −8 $mm$, L 2.6 $mm$, H 6–7 for DRN and H 7–9 for MRN; from Bregma). Micropipettes were lowered at an angle of 26° for DRN and 20° for MRN to avoid the sagittal vein. Extracellular neuronal recordings were carried out using standard procedures with glass micropipette of 10–20 $M\Omega$, filled with 2M NaCl [65–68] or with 2% Neurobiotin (Nb, Vector Laboratories) in 0.5M NaCl solution (14 rats) [69]. Two screw electrodes were placed in the frontal and parietal cortices, and two twisted nichrome electrodes were placed in the dorsal hippocampus (coordinates: AP, -4.4 $mm$; L, -2.6 $mm$; H, 3.4 $mm$) to monitor the electrocorticogram (ECoG).

Neuronal signals were amplified by an AC-coupled amplifier (Dagan 2400A), filtered between 0.3 $Hz$-10 $kHz$ and digitized at 20 $kHz$. Single unit activity was acquired and processed using Spike 2 software (Cambridge Electronic Design, UK). The baseline discharge of raphe neurons was recorded for 3 to 10 min. Neurobiotin was administered by iontophoresis with the following protocol: anodic current, 5 $nA$, 200 $ms$ on / 200 $ms$ off, for 5 minutes [67]. In order to visualize the ECoG, the signal was amplified (×1000), filtered (0.1–100 $Hz$), acquired (512 $Hz$, 16 bits) and processed with Spike 2. The ECoG was used to check the electrocortical state of anesthesia. Only the neurons recorded in the slow wave state or Non-REM urethane [70] were selected for the analyses.

## 4.4 Histological and immunofluorescence procedures

Two types of procedures were performed depending on the experiment. First, in the experiments where the rats did not receive Nb, after finishing the recording experiments and dissecting the brain, it was left immersed in 10% formalin for 48 hours and then was sectioned into slices of 100 $\mu m$ using a vibratome in order to determine the location of the recording electrodes. The path of the micropipette and the location were recognized by light microscopy, and photographs were taken with a histological magnifying glass.

Second, at the end of those experiments in which a neuron was labeled with Nb, the animals were perfused transcardially with NaCl 0.9% heparinized and then 4% paraformaldehyde (PFA). The brains were removed and post-fixation was performed in 4% PFA for 24 hours. Afterwards they were left in immersion in 30% sucrose for 48 hours. They were finally cut into blocks and frozen on dry ice. Coronal sections 30 $\mu m$ of thickness were obtained by a cryostat and stored with cryoprotection solution at -20°$C$.

The identification of the neuron labeled with Nb was performed by two immunohistochemical procedures. In the first procedure, after washing the sections with phosphate buffered saline (PBS), neurons were incubated with PBS plus 0.3% Triton X-100 (PBST) for 90 min. Then, they were incubated with 1% $H_2O_2$ for 30 min. After washing with PBS, the sections were incubated with peroxidase-avidin-biotin complex (ABC 1: 200, Vector Labs) for 120 min and then were exposed to diaminobenzidine (DAB, 0.02%) for 10 min. Then, the sections were washed again and were mounted and coated with glycerol. Finally, photomicrographs were taken in light microscope (Olympus model) to visualize the Nb labeled neuron. In the second procedure, an immunofluorescence was performed. The sections were incubated in 0.5% sodium borohydride for 25 min; thereafter, they were incubated in PBS. Then, they were incubated with streptavidin-Alexa Fluor 555 conjugate (1:5000, Molecular Probes) in PBST 0.3% for 2.5 h. Finally sections were mounted and observed under an epifluorescence microscope in order to find the neuron labeled with Nb (examples are shown in Fig 1). On occasion, more

than one neuron was labeled with Nb, but the location of the recorded neuron was clearly identified.

## 4.5 Data analysis

We sorted the single units according to amplitude and waveform criteria. We examined the lack of spikes during the refractory period ($<2$ *ms*), confirming the absence of contamination by other units. Then, we obtained the Inter-Spike Interval (ISI) sequence. Initially, the ISI sequence from the different neurons had different lengths, so we made their length equal to the smallest number of ISIs registered: $N = 225$. To make a surrogate analysis of the ISI sequences, we randomly shuffled the ISI values to create 100 different realizations.

*Classic measures.* The action potential (AP) of each neuron were averaged and analyzed in shape and duration. The APs were mostly triphasic, and the duration of the first two phases was considered as the AP duration (APD) [32]. The frequency of spontaneous activity (FR) and standard deviation was also calculated. Additionally, the pattern of spontaneous discharge with interval (IH) and autocorrelation histograms (ACH) were analyzed. The coefficient of variation (CV) was used to determine the regularity of the discharge frequency. These analyses were performed in windows of 60–300 *s* of stable activity. Neurons with burst activity were identified by looking at the raw recordings and IH.

*OP Entropy.* The degree of randomness in sets of consecutive inter-spike intervals (ISI) was quantified by the Permutation Entropy [52], which we name OP Entropy, and is the Shannon Entropy [51] of the Ordinal Pattern (OP) encoding of the ISI sequence. Specifically, the OP Entropy is found from $H(S) = -\sum_{\alpha=1}^{D!} p(\alpha) \log_\alpha[p(\alpha)]$, where $p(\alpha)$ is the probability of having the OP symbol $\alpha$ in the encoded ISI sequence (i.e., the relative frequency of appearance of $\alpha$) and there can be $D!$ different OPs. Specifically, OPs are obtained by splitting the ISI sequence into non-overlapping windows with $D$ consecutive data-points and calculating the number of permutations ($\alpha$) needed to organize the ISI magnitudes within each window in increasing order.

We set $D = 3$, so that $D! = 6$. This means that the ISI sequence $\{x(i)\}_{i=1}^{N}$, where $N = 225$ for all neurons and $x(i)$ is the $i$-th ISI value, is divided as: $\{x(1), x(2), x(3)\}, \{x(3), x(4), x(5)\}, \ldots, \{x(N - 2), x(n - 1), x(N)\}$. Then, each window is encoded into an OP, creating a sequence of approximately $N/D = 225/3 = 75$ OPs, $\{\alpha(k)\}_{k=1}^{N/D}$, where each $\alpha(k)$ is the number of permutations needed to order the $k$-th window ($\{x(k), x(k + 1), x(k + 2)\}$) in increasing magnitude. To remove possible cases where $x(i) = x(i + 1)$ for some $i$, a negligible white-noise signal was added to the ISI sequence before performing the OP encoding. Also, to double the number of OPs available to calculate the histograms, a second encoding was done to the ISI sequence, starting from $x(2)$, i.e., $\{x(2), x(3), x(4)\}, \{x(4), x(5), x(6)\}, \ldots$. As a result the statistical power of each $p(\alpha)$ is given by $(2 \times (N/D))/D! = 25$.

*Bins Entropy.* Is the Shannon Entropy [51] of the ISI histogram, which quantifies the degree of randomness in the magnitudes of the ISIs. Its value is obtained from $H_b = -\sum_{i=1}^{N_b} p(i) \log_{N_b}[p(i)]$, where $N_b$ is the number of bins used to construct the histogram. In this work we set to $N_b = 18$ to maintain a similar statistical power as with the OP entropy (i.e., each bin has a priori an average of $N/N_b$ values, which is equal to the $N/D/D!$ bins of the OP encoding).

*Permutation Lempel-Ziv complexity.* Lempel-Ziv complexity (LZC) is an information measure based on the Kolmogorov complexity—the minimal "information" contained in the sequence [53]. This complexity has been used in the analysis of different types of neurophysiological signals, among others for the study of the effects of anesthesia, seizures, and

consciousness [71–73]. To estimate the complexity of a time series $\mathcal{X}(t) \equiv \{x_i; t = 1, \cdots, T\}$ we used the Lempel and Ziv scheme proposed in 1976 [74]. In this approach, a sequence $\mathcal{X}(t)$ is parsed into a number $\mathcal{W}$ of words by considering any sub-sequence that has not yet been encountered as a new word. The Lempel-Ziv complexity $c_{LZ}$ is the minimum number of words $\mathcal{W}$ required to reconstruct the information contained in the original time series. For example, the sequence 100110111001010001011 can be parsed in 7 words: $1 \cdot 0 \cdot 01 \cdot 101 \cdot 1100 \cdot 1010 \cdot 001011$, giving a complexity $c_{LZ} = 7$. A way to apply the Lempel-Ziv algorithm can be found in [75]. The LZC can be normalized based in the length $T$ of the discrete sequence and the alphabet length ($\alpha$) as $\mathcal{C}_{LZ} = c_{LZ}[\log_\alpha T]/T$. Although, initially Lempel and Ziv developed the method for binary sequences, it could be used for any alphabet with finite length. In particular LZC could be applied over the OP discretization, which is known as *Permutation Lempel-Ziv complexity* [76].

## 4.6 Statistics

For the group comparisons shown in Figs 2, 5E, 6B and 7, we used Multilevel Bayesian models, where the null hypothesis was rejected at $p < 0.05$. We chose these models due to the hierarchical nature of the data and bimodal or multimodal and skewed nature of the distributions. A variable number of neurons (from 1 to 7) were recorded per rat, with a total of 169 neurons in 64 rats. The rat was included in the model as a random effect, and the measure to be explored (classic and nonlinear measures) as the fixed effect generating models of 2 levels. All models were estimated using the open-source packages MCMCglmm v.2.30 on R v.3.6.1. The ggforce-package was used for the shaded contours in Fig 3.

For the comparison between measures shown in Fig 4, we used Multilevel Linear models, estimated by the maximum likelihood. F statistic t-test is used to test the relationship between the fixed effects where the null hypothesis is the null model. For these models we also included the rat as a random effect generating hierarchical models with 2 levels. In Fig 4, panels A, B, C, D, F, H, J, K and L, the variables were first transformed to log scale before statistics, since this scale was the likeliest to exhibit a relationship. For these models we used the lmer function from the lme4-package in R. The $r$ correlation was calculated by Pearson's method.

## Supporting information

**S1 Fig. Surrogate analysis of OP Entropy.** Comparison between OP Entropy (left panel) and PLZC (right panel) values from the ISI sequence of all the neurons from the MRN and DRN nuclei (from Fig 2), and the corresponding surrogate values, which are shown in green. OP entropy and PLZC values are significantly different from those obtained by the inter-spike interval surrogates ($p < 0.001$).
(EPS)

**S1 Data. Inter-spike interval data of the MRN and DRN neurons.**
(ZIP)

## Author Contributions

**Conceptualization:** Claudia Pascovich, Nicolás Rubido, Pablo Torterolo.

**Data curation:** Claudia Pascovich, Alejo Rodriguez, Joaquín González, Mayda Rivas, Andrea Devera, Patricia Lagos.

**Formal analysis:** Claudia Pascovich, Diego Serantes, Diego Mateos.

**Funding acquisition:** Pablo Torterolo.

**Methodology:** Claudia Pascovich, Diego Serantes, Diego Mateos, Joaquín González, Diego Gallo.

**Project administration:** Claudia Pascovich.

**Resources:** Pablo Torterolo.

**Software:** Diego Mateos, Nicolás Rubido.

**Supervision:** Nicolás Rubido, Pablo Torterolo.

**Visualization:** Claudia Pascovich.

**Writing – original draft:** Claudia Pascovich.

**Writing – review & editing:** Claudia Pascovich, Diego Serantes, Alejo Rodriguez, Diego Mateos, Joaquín González, Diego Gallo, Mayda Rivas, Andrea Devera, Patricia Lagos, Nicolás Rubido, Pablo Torterolo.

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
