## [Decision Letter · Decision Letter 0]

27 Nov 2023

Dear Dr. Pascovich,

Thank you very much for submitting your manuscript "Dorsal and median raphe neuronal firing dynamics characterized by non-linear measures" for consideration at PLOS Computational Biology.

As with all papers reviewed by the journal, your manuscript was reviewed by members of the editorial board and by several independent reviewers. In light of the reviews (below this email), we would like to invite the resubmission of a significantly-revised version that takes into account the reviewers' comments.

Dear authors, apologies for the delay in the review process. The reviewers have recommended "major revisions" to your manuscript. Please closely follow their suggestions so that your manuscript has the opportunity to be accepted in PLOS Computational Biology.

We cannot make any decision about publication until we have seen the revised manuscript and your response to the reviewers' comments. Your revised manuscript is also likely to be sent to reviewers for further evaluation.

Sincerely,

Claudio R. Mirasso

Guest Editor

PLOS Computational Biology

Thomas Serre

Section Editor

PLOS Computational Biology

Dear authors, apologies for the delay in the review process. The reviewers have recommended "major revisions" to your manuscript. Please closely follow their suggestions so that your manuscript has the opportunity to be accepted in PLOS Computational Biology.

Reviewer's Responses to Questions

**Comments to the Authors:**

Reviewer #1: In this manuscript the authors compare traditional electrophysiological features of neurons with non-linear properties. They analyze neurons recorded from the dorsal and median raphe nuclei, the main sources of serotonergic neurons, and show that some non-linear metrics improve the differentiation between both nuclei.

The quality of the analysis presented in this work is outstanding, and I do believe in the added value of non-linear metrics. However, their utility is not convincing in the present manuscript. The type of neurons compared is unknown. They belong to two different nuclei (DRN and MRN), but they contain a rich variety of neuron subtypes (Huang et al., 2019). Thus, the results cannot be directly associated to the type of neurons but to their location (DRN or MRN). My question is: why is it useful to differentiate between DRN and MRN neurons? A good experimental procedure, as the one presented in this work, would separate them.

As the proportion of serotonergic neurons is much higher in the DRN, the way the manuscript is written suggests that non-linear metrics may identify these neurons, but this cannot be inferred from the analysis and it is not explicitly said (except in section 2.5, but it is another topic). The text revolves around the importance of serotonergic neurons, but most of the results are not linked to them, which is confusing. The relation between non-linear metrics and serotonergic neurons beyond section 2.5 should be clearly stated.

I outline specific concerns below.

1- Goal and limitations of the analysis. As mentioned before, it is not clear the utility of differentiating neurons from two spatially separated nuclei. If the goal is to differentiate serotonergic vs. non-serotonergic neurons, this should be clearly stated in the text. As I think the authors cannot directly relate their results with serotonergic neurons, they should better describe and discuss why they may be related and the limitations of their approach. I

2- The proportion of the main types of neurons in DRN and MRN from previous work should be included and discuss which may be the neurons driving the results.

3- Section 2.1. How the metrics are related is not explained in the methods section. Is it linear correlation? Information about the stats is missing (r2, degrees of freedom…). Which are the best features to differentiate the neurons. Is the inclusion of a second parameter really adding information?

4- What is the percentage of neurons correctly classified as DRN or MRN in the best-case scenario?

5- Figure 4. Difficult to say which are significant from the figure. Maybe include the linear regression only in those that are significant.

6- There are serious limitations in section 2.5. Putative neurons are selected based on linear features and only 7 out of 66 are labeled as serotonergic. (i) Where is the 66 coming from? (ii) The proportion is quite low, suggesting that many serotonergic neurons were incorrectly labeled and non-serotonergic. (iii) The populations are defined by linear metrics, so the differences between them are linear. Indeed, the main non-linear metric separating both groups is the Bins Entropy, which highly correlates with the linear features. (iv) Only two sentences in the discussion about this analysis (293-295), just to mention the result. What is the goal of this section?

7- The Bayes Factor is not explained in the methodology. More information and citations in section 4.4 would be appreciated.

Minor comments:

Line 94: APD of 3.07 +- 0.37ms. Is it mean +- standard error of the mean?

Line 95: ACH and IH should be defined here.

Line 110: Figure 2A (There is only one panel).

Line 294: Figure 6B (It is 7B).

Line 373: Why Nb=18 has the same statistical power as the OP entropy?

Was the multi-unit activity (MUA) excluded from the analysis?

What is the duration of the recordings?

How many rats received Nb?

Reviewer #2: I have reviewed the manuscript titled “Dorsal and median raphe neuronal firing dynamics characterized by non-linear measures” authored by C. Pascovich et al. The authors propose characterizing neuronal activity by means non-linear methods in order to boost the differentiation between functional groups of neurons (Dorsal and median raphe, since they are the main source of serotine in the brain). The main idea is to characterize the spiking dynamics by studying the consecutive inter-spike intervals (ISI).

It is clear that a detailed comparative study between classical and new proposed (non-linear) approaches is presented. Nevertheless, I believe that there are some aspects that have to be addressed.

1.- The ISI time series can be understood as a mapping of the dynamic. That justify the use of a lag = 1 (consecutive ISI). Yet, larger order correlations or characteristics temporal scales can be found when larger lags are used. For instance, I wonder, Does the permutation entropy go to 1 when the lag increases? When analyzing the walk of the ISI sequence, is there self similarity?

2.- I think that a discussion of which dynamical features are found or at least expected to find is missing. The values of the permutation entropy are pretty high, then, the observed dynamic is a stochastic noise? It is temporal correlated? Are there nonlinear features?

3.- In Fig. 4, why to use a linear or a quadratic fitting in the scatter plots between metrics, when is is clear that a more complicated non-linear relation is observed in panels A and J for instance.

4.- I believe that a surrogate analysis is lacking. By comparing the present results against those obtained by a surrogate analysis could help to be more conclusive to the underlying dynamics.

**Have the authors made all data and (if applicable) computational code underlying the findings in their manuscript fully available?**

Reviewer #1: **No: **Data only available upon reasonable request

Reviewer #2: Yes

PLOS authors have the option to publish the peer review history of their article (what does this mean?). If published, this will include your full peer review and any attached files.

Reviewer #1: No

Reviewer #2: No
---

## [Decision Letter · Decision Letter 1]

25 Apr 2024

Dear Dr. Pascovich,

We are pleased to inform you that your manuscript 'Dorsal and median raphe neuronal firing dynamics characterized by nonlinear measures' has been provisionally accepted for publication in PLOS Computational Biology.

Best regards,

Claudio R. Mirasso

Guest Editor

PLOS Computational Biology

Thomas Serre

Section Editor

PLOS Computational Biology

After the revision, the paper can be accepted for publication.

Reviewer's Responses to Questions

**Comments to the Authors:**

Reviewer #1: The authors have done an excellent work addressing all my questions. Now that the goals are clear, the added value of non-linear metrics has been reinforced. I am looking forward their next work.

**Have the authors made all data and (if applicable) computational code underlying the findings in their manuscript fully available?**

Reviewer #1: Yes

PLOS authors have the option to publish the peer review history of their article (what does this mean?). If published, this will include your full peer review and any attached files.

Reviewer #1: No

---

## [Editor Report · Acceptance letter]

10 May 2024

PCOMPBIOL-D-23-01259R1 

Dorsal and median raphe neuronal firing dynamics characterized by nonlinear measures

Dear Dr Rognoni,

I am pleased to inform you that your manuscript has been formally accepted for publication in PLOS Computational Biology. Your manuscript is now with our production department and you will be notified of the publication date in due course.

With kind regards,

Zsofia Freund
